# Ergodic Measure and Potential Control of Anomalous Diffusion

**DOI:** 10.3390/e25071012

**Published:** 2023-06-30

**Authors:** Bao Wen, Ming-Gen Li, Jian Liu, Jing-Dong Bao

**Affiliations:** 1Institutes of Science and Development, Chinese Academy of Sciences, Beijing 100190, China; 2School of Public Policy and Management, University of Chinese Academy of Sciences, Beijing 100049, China; 3Department of Physics, Beijing Normal University, Beijing 100875, China; 4Department of Physics, Beijing Technology and Business University, Beijing 100048, China

**Keywords:** effective ergodicity, anomalous diffusion, convergence rate, friction

## Abstract

In statistical mechanics, the ergodic hypothesis (i.e., the long-time average is the same as the ensemble average) accompanying anomalous diffusion has become a continuous topic of research, being closely related to irreversibility and increasing entropy. While measurement time is finite for a given process, the time average of an observable quantity might be a random variable, whose distribution width narrows with time, and one wonders how long it takes for the convergence rate to become a constant. This is also the premise of ergodic establishment, because the ensemble average is always equal to the constant. We focus on the time-dependent fluctuation width for the time average of both the velocity and kinetic energy of a force-free particle described by the generalized Langevin equation, where the stationary velocity autocorrelation function is considered. Subsequently, the shortest time scale can be estimated for a system transferring from a stationary state to an effective ergodic state. Moreover, a logarithmic spatial potential is used to modulate the processes associated with free ballistic diffusion and the control of diffusion, as well as the minimal realization of the whole power-law regime. The results presented suggest that non-ergodicity mimics the sparseness of the medium and reveals the unique role of logarithmic potential in modulating diffusion behavior.

## 1. Introduction

A current hot topic in statistical mechanics is anomalous diffusion [1]; in essence, the growth over time of the mean squared displacement (MSD) of a force-free particle moving in a medium is not linear but displays a power-law form: $\langle x^{2}\left(t\right)\rangle ∼2D_{\alpha }t^{\alpha }$ (0<α≤2). The exponent α specifies the diffusion process, and the pre-factor $D_{\alpha }$ ensures the correct dimension of such notation. Normal diffusion is observed for α=1. The case 0<α<1 is termed subdiffusion, while α>1 describes superdiffusion. The famous Einstein diffusion formula will no longer apply, so that one obtains the time-dependent diffusion function by calculating the time derivative of MSD and converting it into the form of the generalized Stokes–Einstein formula: $D\left(t\right)=\frac{1}{2}\frac{d}{dt}\langle x^{2}\left(t\right)\rangle =k_{B}T/\left(m\gamma \left(t\right)\right)$ [2,3], where *m* denotes the mass of the particle, $k_{B}$ Boltzmann’s constant, and *T* temperature. Here, γ(t) reduces to a constant only for normal diffusion (α=1). Beyond the Einstein model, a number of physical situations are outside of free Brownian motion and end by surveying nonequilibrium diffusion for a time-periodically driven Brownian particle dwelling randomly in a periodic potential, while constant diffusion coefficients [4,5,6] have also been reported.Nevertheless, for anomalous diffusion, γ(t) tends to infinity for subdiffusion (0<α<1) and decays to zero for both superdiffusion (1<α<2) and ballistic diffusion (α=2). This indicates that the time-dependent diffusion function may only reflect the instantaneous or local diffusion capacity of the particle, as long as the global diffusion linking the different states in the process needs to be studied, a transition problem appears and therefore the estimation of the rate constant is always in question [7].

Recently, some researchers have used the Riemann fractional calculus technique to evaluate the generalized diffusion constant $D_{\alpha }$ [8,9], but this is not proportional to the reciprocal of time and therefore could not be measured in experiments. On the other hand, for only local diffusion or the system to be bound by an external bounded potential, the physical parameters of the system cannot be determined from the time-derivative of MSD, because the latter tends to be a constant. Thus, one cannot obtain useful information in this situation. Furthermore, from the perspective of dynamic processes, friction not only hinders the movement of objects, but also determines the elapsed time required for the system to evolve from an initial to a final state; moreover, the reciprocal of the elapsed time gives an estimate of the magnitude of friction. Calculating the effective friction coefficient for anomalous diffusion is a challenging topic.

To solve the above problem, we use the inverse concept of the famous Lyapunov exponent [10], which provides a measure of the large deviation in the behavior of nonlinear dynamical systems. The reason why two initially infinitesimally close trajectories deviate after a long time is their inherent randomness. When dealing with such problems, the assumption of a finite time scale is often possible. For the Lyapunov index d(t), this can take the form $d\left(t\right)/d_{\infty }=F\left(td_{\infty }\right)$; here, *F* denotes a universal function and $d_{\infty }$ a constant. Although divergence (in chaos) and convergence (in diffusion) are inverse concepts, they both have a common rate of progress; we can use the ergodic convergence rate in instances of anomalous diffusion. Originally developed in the phase transition from an ultracold liquid to glass, it determines the time scale for ergodicity [11,12,13]. As is well known, in nonlinear chaotic dynamics, with two trajectories with very close initial conditions, due to inherent randomness, there is a scale function to characterize the degree to which the two trajectories deviate over time. In diffusion dynamics, however, we consider the opposite situation, a finite-time average of particle velocity or velocity squared, where their distribution width converges to a linear scale function, the shortest time required. To do this, we choose the equilibrium distribution as the initial velocity preparation of the particle.

In a broader sense, we study the time scale required for an anomalous diffusing system transitioning from a state of stationary velocity correlation to its vanishing or rest state, in which the VACF is a time translation invariant. The rest of this paper is organized as follows: In Section II, we propose a squared velocity measure to study the convergence rate of ergodicity and find a scaling law that is generally applicable to anomalous diffusion, from which an effective friction coefficient can be extracted. Several straightforward examples are used to illustrate this approach. In Section III, we use a logarithmic spatial potential to control anomalous diffusion, with the diffusion exponent falling continuously from α=2 to α=0. Section IV presents a study of the VACF for a particle driven by a broadband thermal noise. We demonstrate, here, that the presented approach still works even if ergodicity is broken. Section V presents our conclusions.

## 2. Ergodic Convergence Properties

### 2.1. Finite-Time Scale Function

As is well-known, the ergodic hypothesis is the only fundamental assumption of equilibrium statistical physics; it implies that the long-time average of an observable is equal to the ensemble average; expressly, $\bar{A}=\langle A\rangle$. So far, some necessary conditions for ergodicity, such as irreversibility and the progressive state independence of the initial conditions have been proposed, among which the most attention-grabbing result is Khinchin’s theorem [14]. The observable autocorrelation function gradually vanishes, but it is only suitable for stationary processes, specifically, those for which the autocorrelation function of an observable is invariant under time translation. Generalizations of the divergent Lévy statistics [15] and the aging process [16] have also been completed. The recent literature [17,18] has established the infinite entropy theory, to expand the equality of two averages, while maintaining a constant ratio between the two.

Some information of a physical property (such as the effective friction coefficient) is very important if it can be determined from the history of various states. We consider the finite-time average of observables, i.e., $\bar{A}=t^{-1}\int_{0}^{t}A\left(x\left(t^{\prime }\right),v\left(t^{\prime }\right)\right)dt^{\prime }$ as a random variable and examine its fluctuation width: $\Omega_{A}\left(t\right)=\langle {\left(\bar{A}-\langle \bar{A}\rangle \right)}^{2}\rangle$, defined by the average in the formula $\langle A\rangle =\int A\left(x,v\right)W_{st}\left(x,v\right)dxdv$. If $\Omega_{A}\left(t\right)\to 0$ when t→∞, the system is ergodic [19,20]. A parameter associated with ergodicity breaking in single-particle diffusivity was similarly proposed to demonstrate this condition [21,22].

To extract the transition rate or the effective friction coefficient under the necessary conditions of ergodicity, we define the following time function:(1)$$\begin{array}{c} F_{A}\left(t\right)=\frac{\Omega_{A}\left(0\right)}{\Omega_{A}\left(t\right)}, \end{array}$$
where the fluctuation width for the finite-time average of observable *A* is evaluated using
(2)$$\begin{array}{c} \Omega_{A}\left(t\right)=\frac{1}{t^{2}}\int_{0}^{t}\int_{0}^{t}\left[\langle A\left(t_{1}\right)A\left(t_{2}\right)\rangle -\langle A\left(t_{1}\right)\rangle \langle A\left(t_{2}\right)\rangle \right]dt_{1}dt_{2}. \end{array}$$
Here, $\Omega_{A}\left(0\right)$ denotes the equilibrium value, which is determined by applying l’Hôpital’s rule to the above equation. Although the time average of the observed quantity is not equal to the ensemble average, the question now becomes: How much time has elapsed when $F_{A}\left(t\right)$ becomes a universal function (i.e., linear function) of time? We call this state of the system the efficient ergodic state. For diffusion dynamics, the position of the force-free particle diverges, whereas the velocity is often finite; also, even with the effect of nonlinear bounded potentials, the particle velocity follows a Gaussian distribution. Therefore, measuring and calculating the correlation and fluctuation of particle velocity is very meaningful.

To clarify this problem, we use the GLE model to describe stochastic processes, i.e.,
(3)$$\begin{array}{c} m\dot{v}+m\int_{0}^{t}\Gamma \left(t-t^{\prime }\right)v\left(t^{\prime }\right)dt^{\prime }=\varepsilon \left(t\right), \end{array}$$
where Γ(t) denotes the memory function, which is associated with zero-mean colored noise ε(t). The noise correlation function is linked to the fluctuation dissipation theorem (FDT): $\langle \varepsilon \left(t\right)\varepsilon \left(t^{\prime }\right)\rangle =k_{B}T\Gamma (|t-t^{\prime }\left|\right)$. We apply the Laplace transform method to give the solution in the form of Equation (Equation 3), i.e., $v\left(t\right)=v\left(0\right)h\left(t\right)+m^{-1}\int_{0}^{t}h\left(t-t^{\prime }\right)\varepsilon \left(t^{\prime }\right)dt^{\prime }$ where h(t) is the velocity response function, its Laplace transform is given by $1/(s+\hat{\Gamma }\left(s\right))$. Multiplying the velocity of the two moments and then performing the ensemble average, we obtain a general expression for the VACF,
(4)$$\begin{array}{ccc} \langle v\left(t_{1}\right)v\left(t_{2}\right)\rangle & = & \frac{k_{B}T}{m}h(|t_{1}-t_{2}\left|\right)+\left(\left\{v^{2}\left(0\right)\right\}-\frac{k_{B}T}{m}\right)h\left(t_{1}\right)h\left(t_{2}\right). \end{array}$$
Here, {⋯} the initial average [23]. As the two-time VACF is only a function of the time difference, aging effects vanish, and we say that the system enters a stationary time zone. After a long period of elapsed time, the VACF decays to a constant or vanishes.

For stationary VACF cases, we mainly take the squared velocity as a measure; the calculation of Equation (Equation 2) can be simplified [19,24], and hence Equation (Equation 1) is formally written as
(5)$$\begin{array}{ccc} F_{v^{2}}\left(t\right) & = & \frac{t}{2\int_{0}^{t}\left(1-\tau /t\right)h^{2}\left(\tau \right)d\tau }. \end{array}$$
Importantly, $F_{A}\left(t\right)\to \lambda_{A}t$ should emerge at late times. Here, $\lambda_{A}$ is indeed the ergodic rate of convergence, which also measures the effective coefficient of friction $\gamma_{A}^{eff}$ of the system. So far, a system is said to have effective ergodicity, meaning after a finite time interval, Equation (Equation 5) becomes a linear function of time; it thus has equivalent time average and ensemble average characteristics. This has operational implications, because experiments are usually concerned with time averages, although theoretically ensemble averages are often calculated. Indeed, the value of $\lambda_{A}^{-1}$ represents the shortest time scale for the system to reach the effective ergodicity state [11,13]. For comparison, we consider the velocity itself as another measure, for which the scale function is given by
(6)$$\begin{array}{ccc} F_{v}\left(t\right) & = & \frac{t}{2\int_{0}^{t}\left(1-\tau /t\right)h\left(\tau \right)d\tau }. \end{array}$$

### 2.2. Normal Diffusive Case

To gain an understanding of the present fluctuation measure, we look at the standard Brownian motion as an example: $h\left(t\right)=\exp\left(-\gamma_{0}t\right)$, where $\gamma_{0}$ is the Markovian coefficient of friction. Choosing A=v first, we have
(7)$$\begin{array}{ccc} \Omega_{v}\left(t\right) & = & \frac{1}{t^{2}}\int_{0}^{t}\int_{0}^{t}dt_{1}dt_{2}\left[\langle v\left(t_{1}\right)v\left(t_{2}\right)\rangle -\langle v\left(t_{1}\right)\rangle \langle v\left(t_{2}\right)\rangle \right]\\ & = & \frac{2k_{B}T}{m\gamma_{0}t^{2}}\left[t+\frac{1}{\gamma_{0}}\left(e^{-\gamma_{0}t}-1\right)\right]. \end{array}$$
Using L’Hôpital’s rule, $\Omega_{v}\left(0\right)=k_{B}T/m$, we obtain
(8)$$\begin{array}{c} F_{v}\left(t\right)=\frac{\Omega_{v}\left(0\right)}{\Omega_{v}\left(t\right)}=\frac{\gamma_{0}}{2}\frac{t}{1+\frac{1}{\gamma_{0}t}\left(e^{-\gamma_{0}t}-1\right)}\to \frac{\gamma_{0}}{2}t. \end{array}$$

Next, we select $A=v^{2}$ and use the relationship between the higher-order correlations of the Gaussian variable and its quadratic correlation function, namely, $\langle v^{2}\left(t_{1}\right)v^{2}\left(t_{2}\right)\rangle =2{\langle v\left(t_{1}\right)v\left(t_{2}\right)\rangle }^{2}+\langle v^{2}\left(t_{1}\right)\rangle \langle v^{2}\left(t_{2}\right)\rangle$. We have
(9)$$\begin{array}{ccc} \Omega_{v^{2}}\left(t\right) & = & \frac{1}{t^{2}}\int_{0}^{t}\int_{0}^{t}dt_{1}dt_{2}\left[\langle v^{2}\left(t_{1}\right)v^{2}\left(t_{2}\right)\rangle -\langle v^{2}\left(t_{1}\right)\rangle \langle v^{2}\left(t_{1}\right)\rangle \right]\\ & = & \frac{2{\left(k_{B}T\right)}^{2}}{m^{2}\gamma_{0}t^{2}}\left[t+\frac{1}{2\gamma_{0}}\left(e^{-2\gamma_{0}t}-1\right)\right]. \end{array}$$
Similarly, $\Omega_{v^{2}}\left(0\right)=2{\left(k_{B}T/m\right)}^{2}$, and thus
(10)$$\begin{array}{c} F_{v^{2}}\left(t\right)=\frac{\Omega_{v^{2}}\left(0\right)}{\Omega_{v^{2}}\left(t\right)}=\gamma_{0}t\frac{1}{1+\frac{1}{2\gamma_{0}t}\left(e^{-2\gamma_{0}t}-1\right)}\to \gamma_{0}t. \end{array}$$
Note that $\gamma_{0}^{-1}=\lim_{t\to \infty }\int_{0}^{t}\langle v\left(t\right)v\left(t+\tau \right)\rangle /\langle v^{2}\left(t\right)\rangle d\tau$. This demonstrates that the measure for the squared velocity (proportional to the kinetic energy) gives the same result as the existing theory for standard Brownian motion.

Moreover, if the memory kernel in Equation (Equation 3) is an exponential function: $\Gamma \left(t\right)=\gamma_{0}\tau_{c}^{-1}\exp\left(-t/\tau_{c}\right)$, the VACF is given by
(11)$$\begin{array}{c} h\left(t\right)=\frac{1+z_{1}\tau_{c}}{1+2z_{1}\tau_{c}}\exp\left(z_{1}t\right)+\frac{1+z_{2}\tau_{c}}{1+2z_{2}\tau_{c}}\exp\left(z_{2}t\right), \end{array}$$
where $z_{1}$ and $z_{2}$ are two pole points of the expression $\hat{h}\left(z\right)=1/\left[z+\gamma_{0}/\left(1+z\tau_{c}\right)\right]$. The solution yields $z_{1,2}=\left(-1\pm \sqrt{1-4\gamma_{0}\tau_{c}}\right)/\left(2\tau_{c}\right)$, both of which are negative. We find the asymptotical results of Equations (5) and (6) to be
(12)$$\begin{array}{c} F_{v^{2}}\left(t\right)\to \frac{\gamma_{0}}{1+\gamma_{0}\tau_{c}}t,\;\;F_{v}\left(t\right)\to \frac{\gamma_{0}}{2}t. \end{array}$$
The former is reasonable in physics because, when $\tau_{c}\to \infty$, both friction and noise vanish. The particle undergoes actively free motion, with no transitive process occurring, such that the effective friction vanishes. Nevertheless, the latter result is independent of the correlation features associated with driving noise, and hence the underlying process is suppressed.

Let us provide some additional examples and the most important features compared to those of existing methodologies, in order to offer the most accessible way to understand the advantages of the proposed scheme.In order to analyze frictional aspects of interest for a system moving in a medium, we can see from Equation (Equation 12) that the kinetic energy measurements show more reasonable results than the velocity itself. The rate of equipartition of the kinetic energy is at seven temperatures for the S-peptide and the RNase A enzyme complex [9]. In these results, computed for the peptide and protein in vacuum, at short times, there was a rapid convergence, while at longer times, there was a linear convergence, characteristic of a diffusive process. The rate of kinetic energy equipartition increased linearly with the temperature for the S-peptide and the RNase A enzyme/produce complex. Using a Langevin model, the data were used to estimate the friction constant to be averaged over all atoms; $\gamma_{0}=(0.5$–3) ps ${}^{-1}$ over the range of temperatures studied for both the enzyme/produce complex and the S-peptide. At room temperature, the average velocity relaxation time was found to be $1/\gamma_{0}=0.5$ ps.

### 2.3. Non-Ohmic Memory Case

The key element in GLE is the memory function. One knows that the Wiener–Khinchin theorem relates the noise correlation function to the spectral density of noise. The former is in the time domain and used for calculation, the latter is determined in the spectral space, which reflects the color of noise. Thus, the memory function related to the noise state density (NSD) through FDT is given by
(13)$$\begin{array}{c} \Gamma \left(t\right)=\frac{2}{\pi }\int_{0}^{\infty }\rho \left(\omega \right)\cos\left(\omega t\right)d\omega . \end{array}$$
Once the NSD is known, the GLE describes all types of anomalous diffusion and relaxation within a unified framework. Thereby, both the metrics and scaling function can be analyzed in detail.

Instead, the general non-Ohmic friction model [25,26,27] is described as
(14)$$\begin{array}{c} \rho \left(\omega \right)=\gamma_{\alpha }{\left(\omega /\tilde{\omega }\right)}^{\alpha -1}f\left(\omega \right)\;\;\;\left(0<\alpha <2\right), \end{array}$$
where $\tilde{\omega }$ denotes the reference frequency, f(ω) the frequency modulating function, and constant $\gamma_{\alpha }$ ensures GLE (Equation 3) has the correct dimensional units. Note that the small-ω behavior of the NSD obeys a power law characterized by the exponent α−1. As 0<α<2, one safely sets f(ω)=1 in the theory [26,27]. The Laplace transform of the memory function yields $\hat{\Gamma }\left(s\right)=\omega_{\alpha }^{2-\alpha }s^{\alpha -1}$, where $\omega_{\alpha }^{2-\alpha }=\gamma_{\alpha }{\tilde{\omega }}^{\alpha -1}\sin^{-1}\left(\alpha \pi /2\right)$. At this stage, the velocity response function is expressed by the Mittag–Leffler function, $h\left(t\right)=E_{2-\alpha }\left[-{\left(\omega_{\alpha }t\right)}^{2-\alpha }\right]$ [25,26,27]. For simplicity, we next calculate the reduced VACF numerically by multiplying both sides of Equation (Equation 3) by v(0), and performing the ensemble average using the statistical properties of noise [〈v(0)ε(t)〉=0]. Then, a homogeneous integro-differential equation for the stationary case is obtained,
(15)$$\begin{array}{c} \dot{h}\left(t\right)=-\int_{0}^{t}\Gamma \left(t-t^{\prime }\right)h\left(t^{\prime }\right)dt^{\prime }. \end{array}$$

In Figure 1, we depict the scale function $F_{A}\left(t\right)$ for $A=v,v^{2},v^{3},v^{4}$ as well as various values of the exponent α; however, these models are attenuated by $f\left(\omega \right)=\exp\left(-\omega /\omega_{c}\right)$. All quantities represented hereon in are dimensionless (i.e., we set $k_{B}T=1$, m=1, $\gamma_{\alpha }=1$, and $\tilde{\omega }=1$). In particular, setting α=1 combined with finite $\omega_{c}$ prescribes Ornstein–Uhlenbeck (OU) noise. Furthermore, when $\omega_{c}\to \infty$, Equation (Equation 3) is reduced into the standard Brownian motion, subject to white noise. This is certainly evident from the figure with the numerical calculations, providing quantitative verification of the present analytical results.

Indeed, the effective friction represents the rate of energy exchange of a system within a well-defined environment, with the rate also being inversely proportional to the shortest transitive time for the system, starting from the initial state and finally arriving at the equilibrium state. It is expected that for normal diffusion, the effective friction decreases with the increasing noise correlation time, thus revealing the dependence of the underlying process. The calculated scaling functions (Figure 1) clearly show that the scaling function $F_{A}\left(t\right)$ introduced in this work grows linearly with time after the ergodic time $t_{\mathrm{erg}}∼1/\gamma_{v^{2}}^{\mathrm{eff}}$ for subdiffusion, normal diffusion, and weak super-diffusion. However, our numerical result reveals a prominent trend: for odd moments of velocity (α>1), $F_{A}\left(t\right)$ obeys an increasing power-law in time that is slower than the linear trend; and for even moments of velocity ($\alpha >\frac{3}{2}$), the increase in $F_{A}\left(t\right)$ with time is also slower than that for linearity through to criticality, at which it exhibits ballistic diffusion. A previous work reported the same conclusion from the ergodic properties of the time-averaged MSD, assuming fractional Brownian–Langevin motion [28].

Remarkably, there exists a time scale regime for effective ergodicity, wherein the scaling function $F_{A}\left(t\right)$ of observable *A* assumes linear growth in time. If $F_{A}\left(t\right)$ reaches saturation over time, the ergodicity is broken, and the effective friction vanishes. In contrast to a previous work [29] that claimed “nonergodicity mimics inhomogeneity", our study implies that ergodicity breakdown indicates a thinning of the medium, implying that the particle does not view all parts of phase space.

## 3. Anomalous Diffusion in Logarithmic Potential

Assuming that the system undergoes a non-ergodic process, how does it determine its physical behavior? In this section, we generalize the concept of ergodic convergence and propose a compositional scheme. We assume the memory kernel function in Equation (Equation 3) has the form [30,31,32]:(16)$$\begin{array}{c} \Gamma \left(t\right)=B\left[\frac{1}{\tau_{2}}\exp\left(-\frac{t}{\tau_{2}}\right)-\frac{1}{\tau_{1}}\exp\left(-\frac{t}{\tau_{1}}\right)\right], \end{array}$$
where $B=\gamma_{0}\tau_{1}^{2}/\left(\tau_{1}^{2}-\tau_{2}^{2}\right)$, $\tau_{1}$ and $\tau_{2}$ denote two independent time parameters. Thus, the corresponding ε(t) in GLE is a band-pass noise; when $\tau_{1}\to \infty$, it becomes OU noise, referred to as “red" noise, and when $\tau_{2}\to 0$ and $\tau_{1}$ is finite, ε(t) becomes “green" noise. Furthermore, when $\tau_{2}\to 0$ and $\tau_{1}\to \infty$, white noise is prescribed.

To aid the simulation of anomalous diffusion using the Monte Carlo method, two intermediate variables [30] are introduced to transform GLE (3) with (Equation 16) into a set of Markovian Langevin equations:(17)$$\begin{array}{c} \left\{\begin{array}{c} \dot{x}=v,\\ m\dot{v}=-U^{\prime }\left(x\right)+y_{1}\left(t\right)+y_{2}\left(t\right),\\ {\dot{y}}_{1}=-\frac{1}{\tau_{1}}y_{1}\left(t\right)+\frac{B}{\tau_{1}}v-\frac{1}{\tau_{1}}\xi \left(t\right),\\ {\dot{y}}_{2}=-\frac{1}{\tau_{2}}y_{2}\left(t\right)-\frac{B}{\tau_{2}}v-\frac{1}{\tau_{2}}\xi \left(t\right). \end{array}\right. \end{array}$$
Here, ξ(t) denotes Gaussian white noise, the first two moments of which satisfy 〈ξ(t)〉=0 and $\langle \xi \left(t\right)\xi \left(t^{\prime }\right)\rangle =2\gamma_{0}k_{B}T{\left[\tau_{1}/\left(\tau_{1}-\tau_{2}\right)\right]}^{2}\delta \left(t-t^{\prime }\right)$; moreover, the choice of U(x) is a logarithmic potential [33,34,35]:(18)$$\begin{array}{c} U\left(x\right)=\frac{1}{2}U_{0}\ln\left[1+{\left(x/x_{s}\right)}^{2}\right], \end{array}$$
where $U_{0}$ and $x_{s}$ are two constants.

From the calculated result of MSD $\langle \Delta x^{2}\left(t\right)\rangle =\langle x^{2}\left(t\right)\rangle -{\langle x\left(t\right)\rangle }^{2}$ (Figure 2), two limit cases are analyzed. When $U_{0}=0$, the particle exhibits ballistic diffusion (α=2) [30,36]; that is, the MSD increases with the square of time. If, however, the depth of the logarithmic potential is very large, the motion of the particle is bound, and diffusion saturates after a long time; that is, when $U_{0}\to \infty$, the diffusion power index becomes α=0. More generally, the particle exhibits anomalous diffusion if α∈(0,2); that is, the diffusion index α is a monotonic function of $U_{0}$. Previous studies [37] revealed that the progressive state of particle in a logarithmic potential is a non-Boltzmann distribution, so that the system cannot be “heated”. From the perspective of anomalous diffusion and its control, we also encounter another unique effect when using such potentials.

In Figure 3, we plot the scale functions $F_{A}\left(t\right)$ ($A=v,\;v^{2}$) as a measure of the velocity and its square, respectively. They have a common behavior; i.e., after transient relaxation, both become functions linear in time. While velocity is used as a measure, the coefficient of friction should no longer be determined using the Green–Kubo formula, because the coefficient determined using Equation (Equation 1) is obviously too large for subdiffusion. This is why the denominator of Equation (Equation 6) contains a VACF, which features negative values [30] for sub-diffusion cases; therefore, the value of the denominator becomes too small. Choosing the kinetic energy as a measure has resulted in satisfactory results for other problems [11,13].

We next analyze the results shown in the two insets of Figure 3. When the potential well is very deep, the particle is bound to move and diffusion reaches saturation, so that using the MSD to calculate the coefficient of diffusion and transition time is invalid. Similarly, the measure of the velocity may yield a large amount of friction, as well as a short transition time, because the particle has frozen, without undergoing diffusion. However, if the depth of the potential increases, the effective coefficient of friction is unchanged when the squared velocity is regarded as a measure. Moreover, for shallow logarithmic potentials, the system studied exhibits superdiffusion and its ergodic rate of convergence is slower. Surprisingly, $\gamma_{v}^{eff}$ approaches zero for α>1 and $\gamma_{v^{2}}^{eff}$ becomes very small if $\alpha >\frac{3}{2}$. As anomalous diffusion arises through the velocity response function having a power-law form, we know that $h\left(t\right)∼t^{\alpha -2}$ if $\langle x^{2}\left(t\right)\rangle ∼t^{\alpha }$. Hence, the denominators are given by $a_{1}+c_{1}t^{2\alpha -3}$ in Equation (Equation 5) [38] and $a_{2}+c_{2}t^{\alpha -1}$ in Equation (Equation 6), and their convergence requirements are $\alpha <\frac{3}{2}$ and α<1, respectively.

## 4. Ergodic Measurement

Applying Khinchin’s theorem, we know that the process is ergodic if the VACF tends to zero, otherwise it is non-ergodic. Unfortunately, the rate of transition for the state or the effective coefficient of friction cannot be extracted from the VACF, because its long-time integral tends to infinity for non-ergodic processes. This brings up a problem that has not yet been solved; specifically, although the process is non-ergodic, a determination of the rate of transition between two known states is still necessary, because there is always an associated relaxation process for the evolving system.

We use the Laplace transform method to obtain the analytical solution of Equations (Equation 3) and (Equation 16), the Laplace transform of h(t) being
(19)$$\begin{array}{ccc} \hat{h}\left(s\right) & = & \frac{1}{s+\hat{\Gamma }\left(s\right)}=\frac{\left(1+s\tau_{1}\right)\left(1+s\tau_{2}\right)}{s[\gamma_{0}\tau_{1}^{2}/\left(\tau_{1}+\tau_{2}\right)+\left(1+s\tau_{1}\right)\left(1+s\tau_{2}\right)]}. \end{array}$$
Then, the inverse Laplace transform of $\hat{h}\left(s\right)$ is obtained using the residue theorem,
(20)$$\begin{array}{ccc} h(t) & = & b+\frac{1}{\left(\nu_{1}-\nu_{2}\right)\tau_{1}\tau_{2}}\left[\frac{\left(1+\nu_{1}\tau_{1}\right)\left(1+\nu_{1}\tau_{2}\right)}{\nu_{1}}\exp\left(\nu_{1}t\right)\right.\\ & - & \left.\frac{\left(1+\nu_{2}\tau_{1}\right)\left(1+\nu_{2}\tau_{2}\right)}{\nu_{2}}\exp\left(\nu_{2}t\right)\right], \end{array}$$
where $b={\left[1+\gamma_{0}\tau_{1}^{2}/\left(\tau_{1}+\tau_{2}\right)\right]}^{-1}<1$, $\nu_{1}$ and $\nu_{2}$ are the two roots of the quadratic $\left(\tau_{1}\tau_{2}\right)s^{2}+\left(\tau_{1}+\tau_{2}\right)s+1+\gamma_{0}\tau_{1}^{2}/\left(\tau_{1}+\tau_{2}\right)=0$.

In the literature [30], the above model reveals a limit to thermal diffusion, namely, ballistic diffusion, for which the MSD of the particle increases with the square of time. The cause of this is that the long-term integral over the memory function is equal to zero; i.e., $\int_{0}^{\infty }\Gamma \left(t\right)dt=0$. Obviously, this property is only achieved if there is a negative value in the noise correlation, which means that during the motion of the particle, if the direction of the random force at one moment is positive. Therefore, there is a greater probability of the motion being in the oppositive direction in the next moment. Although this effect is not conducive to large-scale diffusion of a particle, the frictional force experienced during particle motion needs to be considered as well. The combined effect leads to the appearance of ballistic diffusion. As seen from Equation (Equation 20), the related VACF does not vanish asymptotically because h(t→∞)=b≠0; unlike all other diffusions, the velocity of a particle undergoing ballistic diffusion is not an ergodic variable.

Putting $\left\{v^{2}\left(0\right)\right\}=k_{B}T/m$, we know from Equation (Equation 4) that the second moment of velocity is equal to the equilibrium value at any time. However, if either VACF or $\Omega_{v}\left(t\to \infty \right)$ does not equal zero, the system gives arise to non-ergodicity. Our concern is as follows: what is the shortest time the system takes to evolve from state h(0)=1 to state h(t)=b [39]? Furthermore, we define the reciprocal of this time scale as the effective dynamic friction of a non-ergodic system. To this end, we evaluate two time-scale functions
(21)$$\begin{array}{ccc} \gamma_{v}^{eff} & = & \frac{\gamma_{0}}{2}{\left(\frac{1}{\gamma_{0}\tau_{1}}+\frac{\tau_{1}}{\tau_{1}+\tau_{2}}\right)}^{2}, \end{array}$$
(22)$$\begin{array}{ccc} \gamma_{v^{2}}^{eff} & = & \frac{{\left(1+\frac{\gamma_{0}\tau_{1}^{2}}{\tau_{1}+\tau_{2}}\right)}^{3}\left(\tau_{1}+\tau_{2}\right)}{{\left(\frac{\gamma_{0}\tau_{1}^{2}}{\tau_{1}+\tau_{2}}\right)}^{2}\left[\tau_{1}^{2}+\tau_{2}^{2}+\tau_{1}\tau_{2}\left(3+\frac{\gamma_{0}\tau_{1}^{2}}{\tau_{1}+\tau_{2}}\right)\right]}. \end{array}$$

From Equation (Equation 16), we find that when $\tau_{1}\to \infty$ and $\tau_{2}$ takes a finite value, ε(t) is reduced to OU noise, b=0, thus $\gamma_{v}^{eff}=\gamma_{0}/2$ and $\gamma_{v^{2}}^{eff}=\gamma_{0}/\left(1+\gamma_{0}\tau_{2}\right)$. The former is process-independent, the conclusion reached by the usual Green–Kubo formula, and the latter reasonably contains the memory effect. Moreover, if $\tau_{2}\to \infty$, that is, noise and the frictional force vanish simultaneously, the effective coefficient of friction also vanishes. Measuring the squared velocity or kinetic energy of the system is therefore more reasonable.

Plotting the effective coefficient of friction $\gamma_{v^{2}}^{eff}$ against the two parameters $\tau_{1}$ and $\tau_{2}$ (Figure 4), where the squared velocity is measured, shows that the effective coefficient of friction increases with $\tau_{2}$ for a fixed $\tau_{1}$. The monotonic decrease arises because $\tau_{2}$ controls the “red"-ness of noise, and the larger its value, the weaker the noise. Through the fluctuation dissipation theorem, the associated friction also decreases, and even the fluctuations and dissipation vanish. The motion of the particle evolves when conservative forces act, in accordance with theoretical expectations.

## 5. Conclusions

We have established a fluctuation measure of the finite-time average of an observable quantity that worked previously for the phase transition from an ultracold liquid to glass and is generalized to describe anomalous diffusion featuring in the GLE.The measure $F_{A}\left(t\right)$ was found to be a linear function of time at late times for ergodic systems. If the ergodicity of a process is broken, $F_{A}\left(t\right)$ plateaus in the long-time limit. We proved that the present measure of the data shows that different scaling regimes are recovered. The convergence rate for ergodicity is associated with an effective friction coefficient, because the larger the friction, the shorter the transitive time between given states. This indicates that the velocity autocorrelation function must be time translation invariant and decays further to a constant value. Our results demonstrate that the effective friction coefficient for subdiffusion is larger than that of superdiffusion, and thereby, if the ergodicity is broken, it mimics the sparseness of the medium. We have also proposed a minimal and easily controllable model to generate an anomalous diffusion covering all regimes; specifically, a system that undergoes ballistic diffusion in the force-free state falls within a logarithmic spatial potential. The diffusion index can be changed by adjusting the logarithmic potential depth, because the system is subject to a Coulomb-like force. Moreover, fluctuation measures of the kinetic energy should also be used to explore the diffusive dynamics of various systems, such as cells, bound systems, and spin glasses. The presented work holds potential for applications in related research fields.

## Figures and Tables

**Figure 1 entropy-25-01012-f001:**
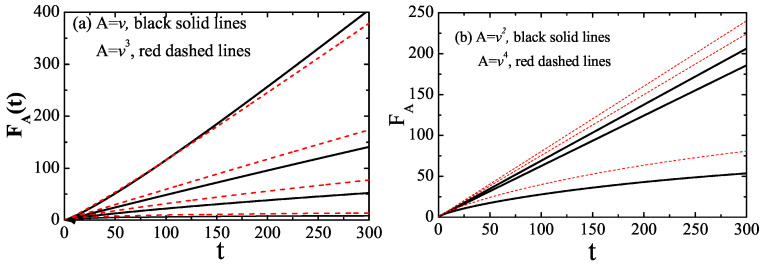
Time dependence of the scaling function $F_{A}$ for the different observables and α, obtained from numerical calculations using Equation (Equation 15) and setting $\omega_{c}=20$: (**a**) $A=v,\;v^{3}$ and α=0.8,1.0,1.2,1.6 from top to bottom; (**b**) $A=v^{2},\;v^{4}$ and α=0.8,1.2,1.6 from top to bottom.

**Figure 2 entropy-25-01012-f002:**
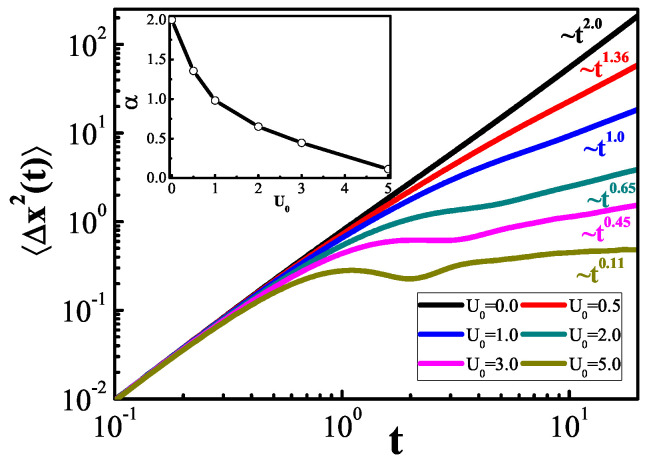
Time variation of MSD for a particle diffusing in logarithmic potentials of various potential depths. Model parameter settings: $\gamma_{0}=1.0$, $x_{s}=1.0$, $k_{B}T=1.0$, $\tau_{1}=1.0$, $\tau_{2}=0.01$.

**Figure 3 entropy-25-01012-f003:**
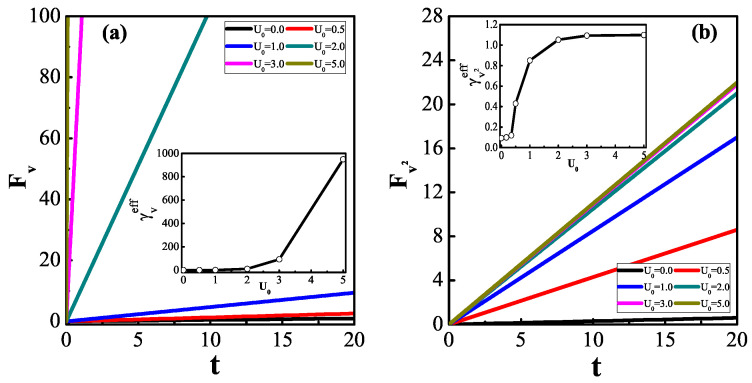
Time variation of scale function $F_{A}\left(t\right)$ (A=v in (**a**), $v^{2}$ in (**b**)) obtained using the same model and parameter settings as in Figure 2.

**Figure 4 entropy-25-01012-f004:**
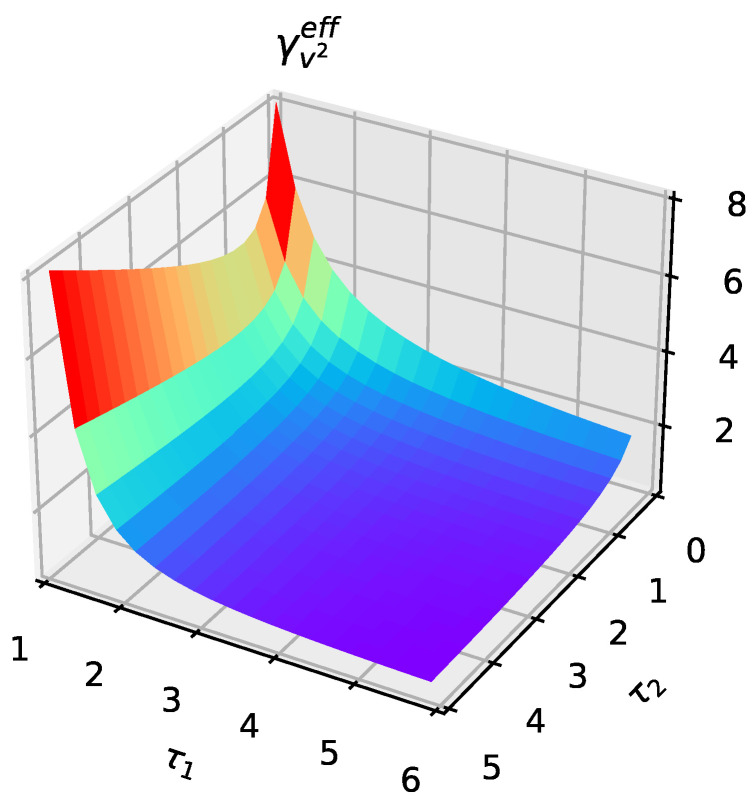
Variation of the effective coefficient of friction with $\tau_{1}$ and $\tau_{2}$.

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
