# Peer review of "Ergodic Measure and Potential Control of Anomalous Diffusion"

_entropy, 2023, doi:10.3390/e25071012_

Round 1
Reviewer 2 Report
1. The analysis on the ergodic measurement is somewhat limited. It would be nice if the authors could provide a more detailed description.
2. The results are also not so indicative. Please provide some examples additional and the most important compared to those of existing methodologies in order to offer the most accessible way to he advantages of the proposed scheme.
3. The Figures must be elevated in quality. May be it is my viewer, yet it would be nice to have a better quality and a more explanatory caption.
4. Are there any restriction in the application of the Wiener–Khinchin theorem? Have the authors something like this? Please define.
Minor errors are to be corrected, but no important for the comprehension of the paper.
Reviewer 3 Report
This paper is a study of the ergodic convergence in anomalous diffusion. I found the paper to be written in a dense form that makes it inaccessible to most readers. The authors assume the reader is steeped in the details of the problem under consideration. I will give examples below.
1. The factor alpha, which appears in an in-line equation on line 16 of the Introduction, is never introduced, let alone explained.
2. The terms subdiffusion and superdiffusion (lines 19-20) are not explained, nor is any source cited for readers who need more information.
3. Line 21-22: "This indicates that, as long as different given states are connected through evolution, a rate of transition problem appears and therefore the estimation of the rate constant is always in question"
What does "different given states are connected through evolution" mean? Why does it follow that "the estimation of the rate constant is always in question"?
4. Lines 24-26: "Of course, this has some theoretical significance because the corresponding coefficient of viscosity does not have the dimensions of inverse time as in the Stokes–Einstein formula and therefore not measured in experiments." I don't understand the meaning of this sentence: What is "coefficient of viscosity"? What are the normal dimensions of that coefficient? Why can't it be measured?
5. Lines 63-64: With reference to the difference between the time and ensemble average of A the authors state: "However, dealing with this measure is difficult because one needs to look for equivalent schemes". What do the authors mean by "equivalent schemes"? How is this a difficulty?
6. Line 99: "For stationary VACF cases, we mainly take the squared velocity as a measure" As a measure of *what*?
7. Line 106 "So far, a system is said to be effective ergodicity". Possibly a grammatical issue, but do the authors mean "to obey" instead of "to be"?
I am not in position to judge the rigor of the treatment in large part because I found the presentation to be impenetrable. I believe that most readers of Entropy will have the same difficulty. I very strongly recommend a major revision that should include a Introduction of the problem in sufficient detail, and of the method of approach. The Abstract should also be clear as to the contribution that this work makes to the state of the art. In its present form it merely announces "an alternative scheme" with no mention of what this scheme is.
English is mostly fine but presentation is not clear
Round 2
Reviewer 1 Report
The authors have taken into account my comments and therefore I recommend publication in the present form.
Quality of English language should be carefully revised in the process of publication production.
Reviewer 2 Report
The authors have indeed elevated their work and conducted the majoirty of my comments.
The language is acceptable.